# Development of Key Principles and Best Practices for Co-Design in Health with First Nations Australians

**DOI:** 10.3390/ijerph20010147

**Published:** 2022-12-22

**Authors:** Kate Anderson, Alana Gall, Tamara Butler, Khwanruethai Ngampromwongse, Debra Hector, Scott Turnbull, Kerri Lucas, Caroline Nehill, Anna Boltong, Dorothy Keefe, Gail Garvey

**Affiliations:** 1School of Public Health, Faculty of Medicine, The University of Queensland, Herston 4006, Australia; 2National Centre for Naturopathic Medicine, Faculty of Health, Southern Cross University, Lismore 2480, Australia; 3Cancer Australia, Sydney 2010, Australia; 4Kirby Institute, UNSW Medicine, The University of New South Wales, Kensington 2052, Australia

**Keywords:** First Nations peoples, Aboriginal and Torres Strait Islander people, co-design, participatory action research, cancer, community engagement

## Abstract

Background: While co-design offers potential for equitably engaging First Nations Australians in findings solutions to redressing prevailing disparities, appropriate applications of co-design must align with First Nations Australians’ culture, values, and worldviews. To achieve this, robust, culturally grounded, and First Nations-determined principles and practices to guide co-design approaches are required. Aims: This project aimed to develop a set of key principles and best practices for co-design in health with First Nations Australians. Methods: A First Nations Australian co-led team conducted a series of Online Yarning Circles (OYC) and individual Yarns with key stakeholders to guide development of key principles and best practice approaches for co-design with First Nations Australians. The Yarns were informed by the findings of a recently conducted comprehensive review, and a Collaborative Yarning Methodology was used to iteratively develop the principles and practices. Results: A total of 25 stakeholders participated in the Yarns, with 72% identifying as First Nations Australian. Analysis led to a set of six key principles and twenty-seven associated best practices for co-design in health with First Nations Australians. The principles were: First Nations leadership; Culturally grounded approach; Respect; Benefit to community; Inclusive partnerships; and Transparency and evaluation. Conclusions: Together, these principles and practices provide a valuable starting point for the future development of guidelines, toolkits, reporting standards, and evaluation criteria to guide applications of co-design with First Nations Australians.

## 1. Introduction

Australian First Nations peoples (hereafter referred to as First Nations Australians) are the oldest, continuous, culture in the world, and are made up of many different cultural groups, with diverse languages, practices, and beliefs. Despite the resilience and cultural strength of First Nations peoples, the devastating burdens of colonisation, marginalisation, social inequality, and racism have greatly impacted their health and wellbeing [1,2,3,4]. The Australian Government’s 2020 National Agreement on Closing the Gap report highlights that First Nations-led approaches to finding solutions to these persistent inequities are urgently needed to make certain that First Nations peoples’ priorities, cultural views and values are centred [3].

Ensuring that First Nations Australians are centred in directing and guiding any endeavours to design and implement solutions to issues that they identify as negatively affecting their community, requires culturally acceptable and effective methodological approaches [3]. Co-design is one approach that is increasingly being used in this regard, both with First Nations Australians and other marginalised groups, due to its potential for empowerment and for prioritising the voices and lived experiences of priority and marginalised populations to self-determine and drive the agenda and find effective solutions to the issues that they regard as important [5,6,7]. While co-design has the potential to be an acceptable methodological approach to redressing the disparities facing First Nations Australians, its application must align with First Nations Australians’ culture, values, and worldview, and privilege First Nations knowledges, expertise, and ways of doing [8].

Our team recently conducted a comprehensive review as part of a larger study commissioned by Cancer Australia, Australia’s peak national cancer control agency that aims to benefit all Australians affected by cancer, and their families and carers. This review aimed to synthesise and describe published evidence around the optimal approaches to co-design in health with First Nations Australians, with application to the cancer control context [9]. Our review, which included 99 peer-reviewed and grey literature documents, identified the following six key themes around optimal approaches to co-design: Aboriginal and Torres Strait Islander leadership; Culturally grounded approach; Respect; Benefit to community; Inclusive partnerships; and Evidence based approach [9]. While the findings of this review provide a valuable starting point to guide co-design approaches with First Nations Australians, a need remained for consideration, deliberation, and review of these findings by a community representation of First Nations Australians to translate these findings into a set of key principles and best practices that are appropriate and effective for guiding applications of co-design in health with First Nations Australians.

Cancer Australia commissioned researchers at the University of Queensland to conduct a study to develop a set of key principles and best practices of co-design in health with First Nations Australians. The aim of this study was to take the findings of the thematic literature review as a foundation for the collaborative development of key principles and best practices of co-design with First Nations Australians that would appropriately and effectively guide the development of health policy, practice, and research. This paper describes the development process and presents the final set of key principles and best practices for co-design in health with First Nations Australians.

## 2. Materials and Methods

To gather the preliminary data to commence this process, our team undertook a comprehensive review of optimal approaches to co-design in health with First Nations Australians, completed in February 2022 [9]. The findings from the thematic analysis of the included literature were used as a starting point to conduct a qualitative stakeholder consultation study to develop a set of key principles and best practices to co-design in health with First Nations Australians. This paper describes the development of these key principles and best practices.

### 2.1. Indigenist Methodology

The University of Queensland researcher team has extensive experience in using Indigenist research methods [10,11] and conducting ‘research at the interface’ [5] to ensure that our work contributes to improving the outcomes and experiences of First Nations Australians. Our approach was guided by the following principles: ensuring First Nations voices and perspectives are prioritised and privileged throughout all aspects of the research process and our work; building First Nations Australians’ research capacity and developing future research leaders; and facilitating collaboration through engaging and connecting with a range of key stakeholders. These principles are congruent with those set out in Cancer Australia’s National Aboriginal and Torres Strait Islander Cancer Framework [4]: First Nations people are engaged and involved throughout the planning, design and delivery of cancer services and research; patients, families, carers, and communities are informed and empowered; working together towards a common goal; and policy and practice informed by reliable data and evidence about what works.

### 2.2. Research Team

Our team acknowledges the importance of reflexively considering and describing our own backgrounds, perspectives, and values that we each bring to the project [12,13]. The first author (KA) is a non-First Nations Australian senior researcher experienced in conducting collaborative qualitative research with First Nations researchers and communities. The second author (AG) is an Aboriginal early career researcher with a background in First Nations Traditional Medicine, health and wellbeing research. The third author (TB) is an Aboriginal mid-career researcher with a keen interest in reducing inequities in health outcomes among First Nations Australians. The fourth author (KN) is a junior First Nations researcher working across the nexus of wellbeing, racism and cancer. The fifth to tenth authors (DH, ST, KL, CN, AB, DK) hold various roles within Cancer Australia, which continues to progress its commitment to First Nations Australian cancer control and to work in partnership to address inequity. The senior author (GG) is a senior Aboriginal researcher with extensive research experience in First Nations Health.

### 2.3. Governance

First Nations governance was provided to the project via two groups: a First Nations Co-Design Working Group (FNCDWG) and Cancer Australia’s Aboriginal and Torres Strait Islander Cancer Control Leadership Group (the Leadership Group). The Leadership Group was formed by Cancer Australia and brings together leaders of influence in Indigenous health, research, and policy as well as consumers affected by cancer. The group champions cross-sector collaboration and leads a shared agenda to improve cancer outcomes at system, service, and community levels in Australia.

The FNCDWG was established and convened to inform and guide the project at critical timepoints. The FNCDWG was comprised of six First Nations people with experience working in research, health care and/or health promotion and policy, and represented a range of experience, age, and backgrounds. The FNCDWG provided important feedback on the emerging findings during the project. Members of the FNCDWG were reimbursed for their time and contributions, as per the Victorian Comprehensive Cancer Centre Alliance best practice consumer and community engagement toolkit and cost model for community sitting fees/hourly rate [14].

Our researchers (TB, KA, GG) presented an online session to the Leadership Group on 3 March 2022 with a summary of the literature review findings [9] and an overview of the proposed approach to stakeholder recruitment. Following this presentation, a Yarning session was held to invite feedback and comment from members of the Leadership Group. The session was recorded and transcribed to enable the inclusion of the feedback provided by members to help inform and guide the analysis.

### 2.4. Ethics

This study received ethics clearances from the University of Queensland Human Research Ethics Committee (Reference: 2021/HE002603).

### 2.5. Stakeholder Mapping and Recruitment

A process of stakeholder mapping was undertaken, which was led by senior First Nations researcher (GG), with support from other First Nations researchers (TB, AG, and a senior researcher at UQ) and a non-Indigenous researcher (KA). Representatives from the following three key stakeholder groups were identified and recruited:First Nations cancer patients, carers and family members;Cancer policy experts and health care providers;First Nations researchers with experience in aspects of co-design and/or participatory research approaches.

Recruitment aimed to achieve a maximum variation sample to ensure that experiences were collected from a range of stakeholders from remote, regional, and urban areas, and with a variety of backgrounds and experiences relevant to co-design in health care for First Nations Australians. We sought to include approximately 30 stakeholders across these three key stakeholder groups, with an aim to include a majority of First Nations stakeholders. Stakeholders were initially approached by email either directly or distributed by a stakeholder organisation.

### 2.6. Stakeholder Consultation

#### Online Yarning Circles (OYCs)

The research team used the findings of the comprehensive review [9] to draft a semi-structured discussion guide that stepped stakeholders through the optimal approaches to co-design with First Nations Australians identified in the literature. Initial development of the semi-structured discussion guide also drew on our extensive experience in qualitative Yarning methodology [3,10].

The consultation with key stakeholders identified in the stakeholder mapping was conducted via OYCs and individual Yarn’s where required, which have been used successfully by the research team previously [15]. The OYCs were facilitated by GG and TB, two Aboriginal researchers experienced in Yarning, and were supported by KA a non-Indigenous researcher experienced with Yarning and AG an Aboriginal researcher experienced with Yarning, who also conducted the individual Yarns. The OYCs and individual Yarns were video recorded with participants’ consent and transcribed to assist in the analysis. Stakeholders who were unable to attend any of the Yarning sessions were offered provision of feedback and input in writing via emailed feedback forms.

Four OYCs and one individual Yarn were held via Zoom during March 2022, with each group comprising members of a particular stakeholder sub-group (e.g., First Nations cancer patients, carers, and family members).

As it can be challenging for some individuals to attend OYCs sessions, stakeholders were offered the option of participating in an individual Yarn with an experienced qualitative interviewer from the research team (GG, AG). This allowed consideration of the individual tailoring of the interaction to suit the needs and preferences of the stakeholders.

Prior to the sessions, our team summarised the themes and sub-themes from the thematic analysis of the comprehensive review into a short one-page document, using the theme names as headings. The sub-themes were translated into actionable items beneath the headings. Stakeholders were emailed the one-page summary to review before the session and were encouraged to have it available to review during the session. The Yarns with stakeholders were guided by a Yarning Guide (see Appendix A). The development of the Yarning Guide was informed by guides developed by the research team for use in previous studies with similar methods. During the sessions, the facilitator guided the stakeholders through each of the different themes and discussed each one and the associated actionable items one by one. Stakeholders were prompted to consider and share their views on the findings, including items that are important, those that needed rewording, moving, or deleting, and anything that they felt was missing.

Demographic information was collected from the Yarning participants via an online survey following their participation. The survey was housed on the secure Qualtrics XM platform. Information collected included State/Territory and postcode/town of residence, stakeholder group, Indigenous status, gender, and age group. First Nations cancer patients, carers and family members were asked about the type of cancer diagnosed; health professional/policymaker/researcher participants were asked what sector and role they work in. This information was collected to ensure diversity of perspectives.

Recordings of the Yarns were transcribed. As transcripts were de-identified, participants were not able to review their individual contributions.

### 2.7. Data Analyses

We used Collaborative Yarning Methodology (CYM) [15,16] led by First Nations researchers, which our team had successfully employed previously [17,18]. This method involves a collaborative and iterative approach to analysis that allows for multiple perspectives via the involvement of several First Nations individuals and groups to discuss the findings and work towards a collective understanding that is in accordance with First Nations values, experiences, and priorities (see Figure 1). This was achieved via the research team meeting regularly to read and discuss the transcripts of the stakeholder Yarns and to develop a coding framework to guide coding and analysis of the data. Input and guidance from the FNCDWG was sought to further refine coding framework. Transcripts of the OYCs and individual Yarn were imported into NVivo 12 software [14] and coded by two researchers (KA, AG) using the coding framework to identify key points of feedback and guidance from stakeholders. The resulting points of feedback and guidance identified through the coding process were applied to the original set of themes and sub-themes from the literature review, which were modified, expanded, and reworded in accordance with First Nations values, experiences, and priorities conveyed by stakeholders. The third meeting of the FNCDWG was held to gain feedback and guidance, which fed into the final set of key principles and best practices.

## 3. Results

### 3.1. Profile of Participants

Four OYCs and one individual Yarn were conducted with stakeholders who had experience and particular interest in co-design with First Nations Australians. A total of 25 stakeholders participated in OYCs and individual Yarns. No stakeholders provided their feedback via written response. The characteristics of participants are described in Table 1.

### 3.2. Qualitative Results

Our iterative analysis of the qualitative data from stakeholder Yarns, with guidance provided from the FNCDWG and the Cancer Australia Leadership Group, identified a set of six key principles and twenty-seven best practices to co-design in health with First Nations Australians, informed and guided by the key themes and sub-themes from the comprehensive review.

The synthesis of the feedback from stakeholders relating to each of the six themes is described in detail below. A summary of each theme and its associated sub-themes, from the comprehensive review, are presented in grey boxes, however, readers are encouraged to refer to the literature review to aid their understanding and interpretation of the feedback from the stakeholder Yarns.

Direct quotes are used throughout the results to illustrate and exemplify the findings; however, quotes are unidentified to safeguard the anonymity of participants.

#### 3.2.1. Stakeholder Feedback on the Theme: First Nations Leadership

First Nations Leadership (as per comprehensive review [9]): This central theme underscores that First Nations Australians and communities must have agency and control over all aspects of the co-design approach and that the processes of co-design must support the empowerment and self-determination of First Nations Australians.

This theme comprised the following three practical sub-themes:-Priorities and processes determined by First Nations communities;-Establishment of governance structures;-Support First Nations leadership.

An overview of the stakeholder feedback on this theme and its sub-themes is described below.

Achieving authentic First Nations leadership is critically important in conducting co-design with First Nations Australians. The language used in this, and other practices associated with First Nations leadership, needs to reflect more strongly that non-Indigenous co-design agents must prioritise and defer to First Nations leadership.

“*Aboriginal people need to have the priority voice, like the non-Indigenous people should be there to support them*.”

“*Sometimes I think [non-Indigenous] people can, and it’s rare perhaps, but people can feel, you may not have to, you know, be subservient. I think for some people, that can be a that can be one of those tensions to navigate.”*

“*It must be maintained and sustained throughout the entire process that non-Indigenous people and organizations take a collegial approach to sharing or deferring leadership*.”

First Nations leadership must not be tokenistic and should ensure the inclusion of people with a range of backgrounds and circumstances relevant to the topic area. The practice Support First Nations Leadership is critically important in co-design and should be listed first in this principle.

“*It’s not just Aboriginal people with a PhD or like an academic job, but there’s community-based leaders and there’s leaders within our health services that have their own knowledge and wisdom, and we should also be recognizing that*.”

“*There are various categories of stakeholders involved and we need to engage the whole hierarchy*.”

The concept of leadership is culturally bound. Western understandings and constructs of leadership are more authoritarian and hierarchical than First Nations understandings, while First Nations conceptions are consensus-driven and relational in nature. A First Nations understanding of leadership should be the model of leadership adopted and implemented in co-design projects, to facilitate culturally appropriate modes of seeking guidance, governance and decision-making.

“*I think leadership is always difficult to define, isn’t it? Because, you know, there’s we have Western concepts of leadership. We also have Aboriginal on concepts of leadership and sometimes leadership is not a word that’s used by mobs, basically because we don’t have that sort of Westminster, you know, legacy of what leadership looks like or those people in control, very different systems*.”

Governance structures must acknowledge and reflect First Nations understandings of leadership and must facilitate broad participation and engagement from all stakeholder groups, rather than a narrow group of individuals. Such structures must be reflective of the community of interest. This will differ between communities, and the most appropriate approach and structure should be determined by the community. Different types of groups that provide leadership and guidance to co-design projects include governance groups, advisory groups, working groups, steering groups and cultural reference groups.

“*A certain organization and certain people, for example, carry a lot of influence in terms of implementation. And that’s why we need to find out, in whichever jurisdiction is engaged in delivering the program, who are the people that needs to be involved and also who are the levels of organization to be approached*.”

“*So that’s about engaging the right stakeholders or a broad range of stakeholders, but also making sure that you have that approval. But because that opens the door I guess for the rest of the community.”*

“*Communities are quite diverse, intricate as well. And so, who do you speak to within community? Just because one family or group may have given approval or support perhaps, they don’t get on so well.*”

“*There needs to be obviously appropriate governance and structures around this to ensure [First Nations leadership] and to facilitate broad participation and not just going to one Aboriginal person and ensuring a broader engagement from all stakeholder groups and that First Nations people are enabled and supported*.”

To achieve robust and appropriate governance, organisations engaging in co-design with First Nations Australians must put aside the prevailing Western dominated leadership models and instead embrace First Nations approaches to decision-making and leadership.

“*So challenging normal Western dominated leadership models to incorporate what Aboriginal leadership is because I think we all understand what Aboriginal leadership looks like, but when you’re negotiating with non-Aboriginal people there’s a lot of blockages and so there’s whether we get that commitment to change and to move and to improve.”*

It must be formally recognised and accounted for in the structure and processes of co-design that First Nations people who are providing leadership have many responsibilities and workloads outside of the project. The provision of their knowledge, time and expertise must not be assumed to be freely available, rather it must be requested, compensated, and recognised.

“*While this is about Aboriginal leadership, quite often when we’re working with perhaps our community controlled organisations or other people, they’ve also got other things to do. So how can we support and embrace that and enable them in leadership around decision making and guiding the project or to service meaningful engagement.*”

It cannot be assumed that all First Nations people and/or organisations who are providing leadership and guidance to a co-design project already have the skills and knowledge required. Support and capability building for First Nations leaders needs to be built into co-design projects. This requires sufficient allocation of time and resources.

“*And so, it’s making sure that there’s support built in around Aboriginal trust on people within communities to enable that leadership and to sort of walk beside them*.”

“*We tend to maybe go in and will do a good job of explaining our particular project and what’s going to happen. And we make this assumption that you know, Aboriginal, the people understand research, the processes, etc. and actually people don’t. And so then they’re working behind the eight ball already expecting to provide this leadership*.”

Seeking appropriate First Nations approvals for the study, which may include Indigenous-specific ethics approvals, community-based approvals, and/or individual consents, are needed to ensure that the processes of the co-design project have been robustly reviewed and vetted by relevant First Nations bodies and people. Moreover, engaging with and respectfully seeking guidance and approvals from community Elders and leaders can be as important as seeking more formal organisational approvals.

“*But unless you’ve got Aboriginal people sitting at decision making tables, you know, in day-to-day operational things, then you know, change what happened. Co-design doesn’t happen.*”

“*We need to call the shots on that we need to establish governance structures. So, who is kind of overseeing this and monitoring it, making sure it’s going okay, who who’s leading it? So that obviously, you know, these co-design things, it’s best if a blackfella is actually leading that because we know our community and how it affects us more so than maybe some non-Indigenous people might know.*”

#### 3.2.2. Stakeholder Feedback on the Theme: Culturally Grounded Approach

Culturally Grounded Approach (as per comprehensive review [9]): This theme conveys that co-design approaches must be grounded within a First Nations worldview, including that all aspects of the approach must centre First Nations voices and values, and the ongoing impacts of colonisation are considered and addressed.

This theme comprised the following four practical sub-themes:-Centre a First Nations worldview;-Account for the continuing impact of colonisation;-Adopt a decolonising methodology;-Strive for cultural rigour.

An overview of the stakeholder feedback on this theme and its sub-themes is described below.

The need to shift to a strengths-based approach that recognises and values First Nations ways of knowing, being and doing is paramount for culturally appropriate approaches to co-design. The language used to describe the practices in this principle is critically important in guiding the approach using a strengths-based lens. Commonly used terms, such as ‘decolonising’, fail to acknowledge and privilege the strength inherent in First Nations cultures and peoples.

“*I think of course we want to adopt a decolonizing methodology, but I think sometimes that takes away from just claiming our own Indigenous knowledge and the way that we work. So, it sort of detracts from it. Let’s claim our Indigenous knowledges and our ways of working rather than focusing on a decolonizing methodology as such.*”

“*I think working towards more of a strength-based approach as opposed to deficit*.”

Acknowledging, identifying, and accounting for prevailing institutional racism is critical for enabling a culturally grounded approach.

“*You said you want to have a strengths-based approach, but I think sort of missing is the big R-word there, which is racism and the acknowledgment of racism within the health system. We’re seeing more of a time because of this systematic racism is still across the whole system*.”

Acknowledging cultural diversity among First Nations Australians and communities must be incorporated into the processes of culturally grounded co-design.

“*The centring of voices from voices and values. That really speaks loudly to me in this particular principle, the Aboriginal Torres Strait Islander voices and values. So, I’m really glad those words are there. But I also think that that it’s important for us to be inclusive of our cultural diversity.*”

The need for achieving cultural competency in a co-design process appears to be misleadingly simple when presented as a single practice.

“*I just have a problem with the term ‘practice cultural safety’, just like some simple thing. Let’s practice cultural safety. So can we do something about broadening that out to what we’re really meaning rather than just something that some of that let’s practice cultural safety has no meaning. But to me it’s just some fairly flippant you know, practice cultural safety, really. What does it mean to me*?”

It must be made clear that cultural competency is not a tick-box or a one-off event. Understanding and attending to foundations of cultural competency is a constant journey, which requires dedicated self-reflexivity and reframing of non-Indigenous people’s perceptions to recognise the prevailing privilege and power differentials, as well as the particular cultural expectations and preferences of the First Nations Australians and communities involved. The complexity and longevity of the process of striving to achieve cultural competency must be reflected in the co-design practices.

“*That’s a really important part of this principle, that non-Indigenous stakeholders do cultural competency training, that they start to learn and understand respectful ways of talking to [community] and working with one another, acknowledging colonised privilege and countering that in order to achieve self-determination, equity, self-determination and equity.*”


*“…it’s one that builds on the cultural competency piece and that, that our non-Indigenous colleagues see that as an ongoing journey. So, it’s not just I’ve done a module in my workplace and now I’m ready to work with [community]. [The principle needs to] say something about it being ongoing and continuing like that.”*



*“So maybe it’s about the basic principles actually defining that cultural practice. Cultural safety in this principle remains to this. I don’t know. Yeah. Okay, well that’s great. We just have to be really concrete on what that means in this in this context. And yes, because I don’t think people understand organizations is still immature in that way to really understand it. They still haven’t really understood what cultural competence is about. They’ve just gotten to the stage of understanding cultural awareness.”*


#### 3.2.3. Stakeholder Feedback on the Theme: Respect

Respect (as per comprehensive review [9]): This theme conveys the importance of grounding the co-design approach in mutual respect, whereby First Nations Australians’ contributions are valued, their competing demands and priorities are respected, and a consideration of culture ensures that all aspects of the co-design approach are accessible and welcoming for First Nations Australians.

This theme comprises the following ten practical sub-themes:-Practice cultural safety;-Embrace flexible and iterative processes;-Allow adequate time and resources;-Acknowledge and respond to First Nations diversity;-Seek appropriate community and ethical approvals;-Establish regular and sustained culturally appropriate communication;-Establish conflict resolution protocols;-Set reasonable expectations;-Provide fair renumeration;-Use First Nations branding and design.

An overview of the stakeholder feedback on this theme and its sub-themes is described below.

The principle of respect is critically important and by its nature is also very broad. While many of the practices listed in this principle are directly related to the enactment of respectful co-design, there are a number of practices that also fit well with other principles. Moving some of these to other principles will make the Respect principle less cumbersome to comprehend and address.

It is important that the expectations placed upon First Nations people and organisations within co-design projects is reasonable and not tokenistic. This needs to encompass both the nature and suitability of the task or role, as well as the amount of time and resources that are required. It is common for First Nations staff within organisations to receive many requests to take on additional work and responsibilities on top of their usual workload in order to bring a First Nations perspective or view. This can result in individuals being overwhelmed by requests and work, which can cause burn out. Moreover, First Nations individuals cannot be reasonably expected to be appropriately placed or experienced to provide representation on First Nations issues.


*“Engagement is continuing. It’s a continual learning. It’s just not a one-off call.”*



*“First Nations participation has to be not tokenistic. It’s trying to think in the opposite way to that, but you know, fully engaged but not, not tokenistic…authentic…”*



*“[communities], they’ve also got other things to do. So how can we support, support and embrace that, and enable them for leadership around decision making and sort of guiding the project or to service meaningful engagement… It’s maintained and sustained throughout the entire process and that non-Indigenous people and organizations also take a collegial approach to sharing or deferring leadership.”*


There is a need to strengthen and formalise mandates to recognise and compensate First Nations people for the time, work and expertise that they put into co-design projects. Community members must never be expected to spend their time and expertise on a project without being fairly compensated. The need for equity in remuneration is critical, which goes beyond ensuring the compensation is adequate and reasonable.


*“One of my bugbears is that government agencies continually expect Aboriginal people to contribute their time without being paid.”*


The issue of respecting the timelines of communities and community members is highlighted as something that is commonly disregarded by non-Indigenous organisations. Timelines must take into account the need for longer discussion and decision-making processes that are common processes for First Nations communities, as well as the other priorities and responsibilities they hold. Estimating the amount of time that is required to build into co-design projects’ timelines, should be determined by and/or in consultation with the communities involved. They are best placed to know local circumstances that will impact timing and likelihoods of tasks being achieved.


*“I do think that there’s still a lot of work to do… to encourage people to feel like it’s okay to take a bit longer time wise. I think still sometimes we as a collective were rushed through processes. Where actually our way is to give time, the time it needs.”*



*“I may have three months to do something. But that’s just me, you know? So, there’s tension there. Do I impose or compel people to do things with within three months with me or do I need to reflect on what it is I’m requesting, asking, seeking to do?”*


The language that is used in co-design projects needs to be accessible, inclusive and relevant to First Nations Australians and communities. Terms like ‘consumer’ are not familiar or welcoming for community members. This is an example of how a commonly used term can denigrate groups within a co-design project and deter First Nations people from engaging with co-design.

#### 3.2.4. Stakeholder Feedback on the Theme: Benefit to Community

Benefit to Community (as per comprehensive review [9]): This theme conveys that the processes and outcomes of the co-design approach ensure meaningful and sustainable benefit to First Nations Australians and communities.

This theme comprises the following three practical sub-themes:-Work to achieve tangible and sustainable positive outcomes;-Formalise First Nations knowledge ownership and sovereignty;-Enhance capabilities of First Nations Australians.

An overview of the stakeholder feedback on this theme and its sub-themes is described below.

The need to ensure that co-design offers benefit to First Nations Australians and communities is an essential yet complex issue. The meaning of ‘benefit to the community’ needs to be spelled out clearly to identify the multiple ways in which this benefit must be realised. Fundamentally, the benefits must be identified and regarded as beneficial by the community themselves, and not presumed by others. In other words, the community themselves must determine the benefits to be gained. The current wording is too flexible in this area and does not provide strong enough directives for the co-design team to engage the community to understand and build into the project the nature, timing and scope of the benefits that must be realised for the community.


*“…people will say “Oh, we’re gonna, you know, generate this knowledge for cancer, and that’s going to be a benefit to community” …how are you going to negotiate those benefits? And what about if community have different perceptions of what the benefit is? How are you going to navigate that? Usually the academics, they’re the ones that say what a benefit is rather than our community.”*



*“A lot of non-Indigenous people have become professors on our disadvantage… I critique them because usually it’s the same thing, “oh, we’re going to benefit to community it’s going to be that we hire an Aboriginal research assistant. Isn’t that going to be great? And they’re going to work for us”. What is that in reality? What are they doing? They’re collecting the data for you so you don’t have to do it. You don’t know Aboriginal people, you wouldn’t get the data if you walked the streets… So of course that’s going to benefit you. But in their eyes “benefit to community. I gave a job to a poor black person”. And then they’re going to take that data and they’re going to write whatever they want and do whatever they want.”*



*“…what is the point of any of this research or work unless it makes a meaningful difference at a grassroots level for our people?”*


There is a pressing need for transparency about what the benefits of the co-design project are for all parties involved. This includes financial, academic, managerial, capability and reputational benefits that non-Indigenous members of the co-design team stand to gain from the project.


*“What [community] are getting out of the partnerships and being clear and upfront around that because, you know, a lot of non-Indigenous researchers are also getting [benefits] out of the research working with community; we need to be upfront and acknowledge that…”*



*“But, you know, when you have a partnership and you work together on a project and then you write it up in a report and say you’ve got a sustained relationship with partners moving forwards in the future, that may only have benefits for one of the partners. So, they keep coming back to that First Nations group and saying, you know, we are consulting, we’re consulting about X, Y and Z with them… we keep going back to the same people and wanting more advice and more partnership, more direction, for what value for [community]? …It’s not mutual benefit.”*


The need for a strengths-based lens and language in this principle and associated practices is important. The phrase ‘enhance capabilities’ implies an existing deficit in capabilities. The capabilities that are being supported should be identified by the community. For the co-design process to truly be of ‘benefit to community’ these capabilities need to be identified by the community themselves. The need for enhanced cultural capabilities in non-Indigenous people and organisations was flagged as being in many ways equally important as building those of First Nations people.

“*You know, if we were to enhance capabilities, what would that look like in that context? Well, that’s the thing that we’d all have to negotiate and work together.”*


*“And I think in the “enhance capabilities for Aboriginal people” we also want people to recognize Aboriginal people have capabilities and sometimes we need to enhance the non-Indigenous or even the academic”*


There is a need for greater clarity around how data sovereignty and ownership of the data are achieved in a co-design project. This will be dependent on the structure and needs of the specific community involved.


*“Firstly, sovereignty of data. In the past, data after it has been collected from the community is sent off and they only hear about it again in published research often presenting them as ‘others’. A keeping place of knowledge would be useful for mob. For example communities with a community research panel.”*


#### 3.2.5. Stakeholder Feedback on the Theme: Inclusive Partnerships

Inclusive partnerships (as per comprehensive review [9]): The quality, strength, and equity of partnerships between stakeholders is paramount. Relationships foster collaboration, two-way learning, and have clear, agreed, and documented processes that ensure transparency and accountability.

This theme comprises the following three practical sub-themes:-Foster a collaborative approach;-Support self-determination and equity for First Nations Australians;-Build sustained relationships;-Ensure transparency and accountability;-Create a shared space for two-way learning.-An overview of the stakeholder feedback on this theme and its sub-themes is described below.

Central to the success of co-design with First Nations Australians is the quality and longevity of meaningful relationships with the community. The process of building these relationships must begin well before embarking on a co-design project. While this may not fit with usual funding cycles and project timelines for non-Indigenous organisations, it is critically important for co-design conceptions to originate from and in partnership with the community.


*“I think I’m always like the old adage of, you know, working with our mob, you know, you do people before you do work. So, you know, taking that time down get to know who you’re working with the community.”*



*“Underpinning that is the respect for relationships and trust and listening to and taking the lead from the First Nations peoples and those involved, rather than just saying, ‘well, okay, I’ve sought ethical approval’ or ‘I’ve ticked these boxes.’ That’s really respecting me, the sort of fabric that should underpin all of that.”*


If key relationships are not in place in the conception phase of a project, there will be no mechanisms for community voices to guide the nature and approach of the project. Furthermore, for First Nations Australians it is the cultural expectation that a level of trust and shared understanding between all parties is developed and achieved before the co-design ‘business’ is commenced.


*“…research isn’t necessarily the first thing that’s done… there’s a need for a period, you know… to get to know each other or to, you know, to break the ice, or to establish relationships in ways that aren’t necessarily research focused.”*



*“…getting to know you face to face might be something to highlight… so rather than worrying about not doing the wrong thing or doing the right thing during the research.”*



*“…people get to know to know each other and my sense is that that can help… But that happens, you know, before the business happens.”*


Achieving equity in the partnerships between stakeholders will involve considerations of existing and prevailing power differentials and dynamics. Taking steps to actively shift these to ensure that the community is in the driving seat is a central point that needs to be emphasised. Moreover, recognising that the community has the lived experience and the local knowledge of the processes, expectations and needs in their area, not only saves time and frustration, it is centrally important to value and prioritise these types of expertise in order to achieve equitable partnerships.


*“You know, a lot of institutions decided that they’ve got power to make decisions for us and things. So making sure that power balance has shifted to the community being in the front seat and driving; that they’ve got the power to make calls.”*



*“So it’s about acknowledging that when you’re coming into this space that those partnerships that you’re working within might have power imbalances historically and in the present day. And so, making sure that the power is with [community] to make those decisions.”*



*“…a partnership with two-way learning is good, but I think we need to recognize that Aboriginal communities have a whole range of capabilities… I think when we look at work like this, best practice is to recognize that Aboriginal and Torres communities have a whole range of capabilities that can benefit the researcher.”*


Communication is a central pillar of relationship building and maintenance. There must be clear and consistent two-way communication between stakeholders, which is inclusive of community voices across the life of the co-design project. Such communication must continue beyond the completion of the project. This approach ensures that everyone involved has a voice and that meaningful partnerships with the community are maintained.


*“…taking the time to have those yarns, if that’s what that community needs... And when you talk to them, and everybody feels important, everybody thinks alright, yeah.”*



*“…with my [communities], I review that every year or two… what do I need to be doing? What do I need to be sharing? How do I need to work with you? but no one actually cares to track it, so they can be like, “Oh no, I sent all of them minutes and the final papers to communities and they’re up to date”, but it’s not meaningful.”*



*“So taking the time to have those yarns, if that’s what that community needs? And when you talk to them and, and everybody feels important, everybody thinks alright, yeah. They’re, actually, genuinely looking after my welfare and I’m participating because I understand how it’s going to benefit me.”*


Embedding effective two-way communication channels throughout the project can foster accountability within the project.


*“…’grassroots accountability’—needs to be a reflexive process with checks along the way.”*


#### 3.2.6. Stakeholder Feedback on the Theme: Evidence-Based Decision Making

Evidence-based decision making (as per comprehensive review [9]): This theme conveys the importance that co-design methods are guided by evidence-based best-practices. This theme comprises the following three practical sub-themes:-Strive for evidence-based rigour;-Build in evaluation and monitoring processes;-Ensure the outcomes are co-designed.

An overview of the stakeholder feedback on this theme and its sub-themes is described below.

The title of this principle is considered too narrow and off topic to effectively and appropriately guide co-design with First Nations Australians. The principle of Transparency and Evaluation better encapsulates the concept. The question around the nature of the evidence that is referenced here is key. Evidence that is valued and relevant to the lives of First Nations Australians should be privileged over Western biomedical constructs and procedures. Orientating this principle towards shared decision-making is a better fit with First Nations values and processes.


*“…the point about ‘evidence-based decision making’, it kind of creates a little bit of tension you know; where ‘shared decision making’ is actually a better statement, because then that could include ‘evidence based’, but it comes from, you know, that negotiation… for and benefit to the community.”*



*“Is ‘evidence-based decision making’ the right name for that... maybe not because I mean to me, it’s like, ‘What? What does that mean?’”*


First Nations ways of knowing, being and doing need to be reflected in this principle. Making it clear that Indigenous knowledges are to be included and respected is important.


*“…how do we acknowledge the Indigenous world view and Indigenous knowledge within ‘evidence based decision making’ in this co-design process? Because I think it’s just as important as what the literature says.”*



*“…it’s not just about ‘evidence based’ from what publications say. It’s about ‘evidence based’ from a community perspective about how things work and the way we should be doing things.”*


There needs to be transparency around the methods used for co-design. This can be achieved through reporting the whole process, so it is clear where parts have been co-deigned and where they have not.


*“I think some way to kind of bring it back to kind of transparency around what is co-design, how you did it. Like I feel like when we talk about like the methods, I think people need to be transparent about their methods.”*



*“…People say, “Oh, well, I co-designed it, then I went off and did this” You didn’t co-design the study, you didn’t co-design the outcome measures, and what data collection tools you were using and all of those things. You co-designed maybe the resources, or a component of a study, rather than a whole study.”*



*“We’ve got to have really clear and agreed and documented processes to ensure that what we’re doing is clear. It’s not being hidden behind anything, and we’re really accountable that way. You know, we’re respectful and responsible to one another in this co-design approach.”*


While the process of co-design itself is important, there needs to be real and tangible positive outcomes for community stemming from the co-design process. This includes a commitment from all partners to follow through to implement the findings or recommendations of the co-design process. The outcomes of the co-design process also need to be measurable either quantitatively or qualitatively to enable evaluation and accountability.


*“We talk, talk, talk, but we don’t actually really develop and do okay… when are we going to actually get to do something like putting in solutions right? So making sure that co-design has a real action, an outcome.”*



*“So, you know, we’ve got so much data on a lot of it. And but I guess it’s just about not reinventing anything and not coming up with something that is not going to be able to be developed… or we keep going around in a circle type of thing.”*



*“…making sure that’s measurable, tangible, like you can touch it and measure it.”*


The importance of what is considered ‘rigor’ cannot be overstated. Decisions that are made about what data to collect, how to collect it, and all other aspects in the methods of co-design, will have an impact in the future for First Nations Australians. These need to be informed by the community, they must be appropriate for the community, only then will the co-design project be truly rigorous.


*“I fight with people all the time “no this is rigorous, this is the most rigorous way to test this” and I say yeah but if it’s not appropriate for Aboriginal people, then it’s not the most rigorous way to test something.”*



*“I’ve had a conversation… where we’re talking about RCTs [Randomised Controlled Trials], and like obviously everyone wants an RCT, and she said ‘oh, I have this beautiful step wedge plan and it’s so beautiful’ and when she went to her community partners, and they said ‘uh uhhhh’... So sometimes you’ve got to pull some of that rigor that you want away because it’s about co-design, it’s about being appropriate for our community.”*


#### 3.2.7. Final Set of Key Principles and Best Practices for Co-Design

The feedback from stakeholders on the themes of the literature review, together with guidance from the FNCDWG and the Cancer Australia Leadership Group, was used by our research team to revise, expand, and refocus these themes into a set of six key principles and 27 related best practices that could be used to guide the application of co-design in health with First Nations Australians. The titles of the six principles remained in line with the themes identified in the comprehensive review, with the exception of Principle 6, which saw the significant refocusing from the comprehensive review theme, Evidence-based decision making, into the principle called Transparency and evaluation. The key principles and best practices are presented in Table 2.

## 4. Discussion

This paper describes a study to develop a set of key principles and best practices to co-design in health with First Nations Australians. The findings of this study identify this set of six key principles, which are accompanied by 27 associated best practice approaches: First Nations leadership, Culturally grounded approach, Respect, Benefit to community, Inclusive partnerships, and Transparency and evaluation. Together, these principles and practices provide a valuable starting point for the future development of guidelines, toolkits, reporting standards, and evaluation criteria to guide applications of co-design with First Nations Australians. Below is an examination of some critical issues that require consideration in the implementation of these principles and practices in a way that supports genuinely equitable partnerships and fosters shared decision-making in co-design with First Nations Australians.

The call for the health policy and practice to be citizen- and/or patient-centred has been growing steadily for decades. The role of co-design as a potential mechanism for the inclusion of patient voices in health care and policy has been broadly identified, with applications evidenced in a large co-design initiative underpinning the National Disability Services in 2015 [19]; the NSW Council of Social Service (NCOSS) promoting the principles of co-design in 2017 [20]; and Australian Department of Health through Primary Health Networks endorsement of a co-design approach in commissioning was proposed for projects funded by the in 2018 [21]. While the patient perspective is acknowledged as a core pillar of safe, high-quality care in Australia’s health system [22,23], there is evidence that the voices of First Nations patients are not always being heard in this regard [24,25,26]. Significant steps are being made at a national level to address this gap, with co-design approaches recently being embedded within the new National Agreement on Closing the Gap [3], and the establishment of the Indigenous Voice to Parliament, which is underpinned by co-design principles [27]. And while these new mechanisms offer promise for First Nations inclusivity and sovereignty, it is important that the increasingly ubiquitous use of co-design terminology reflects a movement towards citizen-centric design, rather than merely a “*shift in terminology, and/or a minor change at the margin*” [5].

In order for co-design approaches with First Nations Australians to be effectively, the evidence this study underscores the need for co-design applications with First Nations Australians to move beyond consultation. The principles identified here highlight the requisite for First Nations Australians to be empowered to drive the agenda, the process and the determination and implementation of the solutions. Achieving this requires the shifting of long-standing power differentials to facilitate equitable and shared decision-making; the achieving of which is undoubtedly the greatest challenge facing governments, institutions, and organisations in creating a space for authentic co-design to be operated. The principles and practices identified in this study provide some valuable guidance in how First Nations Australians want to engage in co-designing solutions to issues they face.

A set of seven potential negative consequences inherent in the application of co-design have been identified by Steen et al. (2018) and further elaborate by Dillon (2021) [5,28]. These potential negative outcomes are: the deliberate rejection of responsibility; failing accountability; rising transaction costs; loss of democracy; reinforced inequalities; implicit demands; and co-destruction of public value [5,28]. Awareness and consideration of these potential negatives is important in countering, what Steen and Dillon regard as, a burgeoning unquestioning optimism around the benefit of co-design in guiding policy and practice [5,28]. These vulnerabilities in the method are noteworthy, and are able to be reduced, if not avoided, with proper adherence to the principles and best practices detailed in this study with any application of co-design with First Nations Australians.

### Strengths and Weaknesses of the Study

A key strength of this study is the strong representation of First Nations Australians authors (TB, AG, GG), who were involved in leading and guiding all aspects of the study. Further, the inclusion of a variety of First Nations stakeholders from across the country is another important strength of this study, as this ensured that a diverse array of perspectives and voices informed the development of the final set of principles and practices. One weakness of the study is that there was a tight timeline allocated to conduct the project, which impacted the number of stakeholders who could be recruited to participate in the Yarning Circles and time spent analysing the data. Building additional time into projects requiring extensive stakeholder recruitment and consultation would benefit future studies conducted in this area. Additionally, it must be acknowledged that the sample of stakeholders who participated in this study included more women than men, which may have had some impact on the findings. While any impact is likely minor, this should be explored and established in future studies.

## 5. Conclusions

The adoption of co-design approaches in health policy and practice contexts continues to proliferate, as its potential value is recognised for engaging First Nations Australians and communities as leaders in the processes of finding solutions to complex issues they face. While there are genuine potential benefits of co-design, the ways in which the methodology is operationalized is centrally important in determining its efficacy in enacting culturally safe and effective co-design and avoiding the potential negative consequences associated with the method. The key principles and best practice approaches to co-design developed in this study offer a constructive guide to commence a process of co-design with First Nations Australians. It is important to note that each application of co-design requires a tailored approach that is grounded in the specific needs, characteristics, values and culture of the people and communities involved. Ensuring transparency in and evaluation of any application of co-design is a critical component of the process to maintain accountability to the First Nations Australians and communities it is intending to serve.

## Figures and Tables

**Figure 1 ijerph-20-00147-f001:**
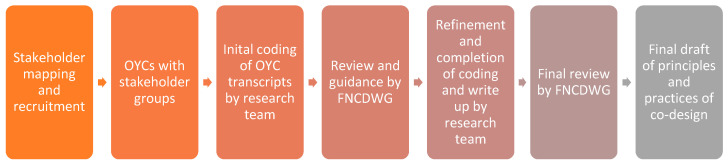
Flowchart of the development process.

**Table 1 ijerph-20-00147-t001:** Stakeholder Demographics $\left(n=25\right)$.

	n	%
Sex		
Male	8	32%
Female	17	68%
Indigenous Identification		
Indigenous	18	72%
Non-Indigenous	7	28%
Stakeholder Group		
Patient	5	20%
Research	10	40%
Policy/Health Service and Provision	10	40%
State		
NSW	11	44%
QLD	5	20%
NT	2	8%
WA	2	8%
VIC	3	12%
SA	2	8%

**Table 2 ijerph-20-00147-t002:** Key Principles and Best Practices for co-design in health with First Nations Australians.

**Principle 1: First Nations leadership** *Principle overview:* Authentic and appropriate First Nations leadership must be embedded within the co-design approach. The nature and function of this leadership must be determined with the community and should reflect the community’s structure and interests. First Nations leadership needs to begin during the conception stages of the project and extend beyond the completion of the project.
Principle 1 Best Practices: 1. First Nations leadership directs all levels and components of the project 2. Priorities and processes are determined by First Nations communities 3. Capabilities that support First Nations leadership are nurtured and strengthened 4. Governance structures are determined by the community and reflect community interests 5. Appropriate cultural and community approvals are obtained
**Principle 2: Culturally grounded approach** *Principle overview:* The strength and diversity of First Nations culture must be reflected in the co-design project and approach. A continuous process of striving for cultural competency and self-reflection by non-Indigenous people involved in the project is essential.
Principle 2 Best Practices: 6. Continuous striving for cultural competency is reflected in the processes 7. Diversity among First Nations Australians and cultures is acknowledged and responded to by the project 8. First Nations worldview underpins all components of the project 9. Strengths of First Nations cultures, knowledges and peoples are recognised and applied throughout the project 10. Continuing impacts of colonisation are acknowledged and mitigated
**Principle 3: Respect** *Principle overview:* The expertise, experience and time of First Nations people and organisations must be respected and recompensed to ensure equity for all involved in the co-design process. Approaches must incorporate flexibility and must be familiar and engaging for First Nations Australians to feel comfortable and welcome to lead and participate in the process.
Principle 3 Best Practices: 11. Expectations of First Nations people and organisations are fair and reasonable 12. Provision of remuneration to First Nations people and organisations equitably compensates their time and knowledge 13. Adequate time and resources are provisioned to successfully complete the project objectives 14. Flexible and iterative processes are embedded and facilitated 15. Culturally appropriate language, branding and design is used throughout the project
**Principle 4: Benefit to community** *Principle overview:* Co-design projects must aim to serve First Nations Australians and communities above all else. The project conception, design, processes and outcomes must provide timely, tangible and sustainable benefits that are valued by the community.
Principle 4 Best Practices: 16. Community sets the agenda for the project 17. Outcomes of the project offer tangible and sustainable benefits that are valued by the community 18. First Nations people and communities own the data and the knowledge resulting from the project 19. Capabilities valued and prioritised by First Nations people are fostered and strengthened
**Principle 5: Inclusive partnerships** *Principle overview:* Fostering and maintaining equitable and collaborative relationships between all participants is central in driving effective co-design projects. Establishing appropriate communication channels and conflict resolution processes, formulated by and with community, that maintains trust and supports authentic partnerships is imperative.
Principle 5 Best Practices: 20. Sustained collaboration that fosters two-way learning is developed and maintained 21. Self-determination of First Nations Australians is supported by the project 22. Achieving equity for First Nations Australians is prioritised 23. Regular and sustained culturally appropriate communication channels are maintained 24. Conflict resolution protocols are formalised to manage disagreement
**Principle 6. Transparency and evaluation** *Principle overview:* Transparency in all aspects of the co-design project is essential. Accountability to First Nations leaders must be formalised and embedded into co-design projects. Embedding monitoring and evaluation throughout the co-design process using agreed performance indicators that facilitate transparency and accountability to community is essential.
Principle 3 Best Practices: 25. Transparency in decision making and benefits to stakeholders is ensured to enable accountability 26. Monitoring and evaluation processes are built into the project 27. Project outcomes are not pre-determined and are authentically co-designed

## Data Availability

Not applicable.

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
