# Peer review of "Development of Key Principles and Best Practices for Co-Design in Health with First Nations Australians"

_ijerph, 2022, doi:10.3390/ijerph20010147_

Round 1
Reviewer 1 Report
I thank the authors for such a fine and valuable research and results. I have a few (minor) suggestions to improve the paper, and I hope to see it publish in the near future. Thank you!
The paper is extremely well written in my view, extremely relevant and timely, and both inspirational and a strong example of how rigorous and cultural appropriate research can be done and reported on. Thank you.
My recommendations pertain mainly to adding clarification in the earliest sections of the paper, given the topic.
First, in the abstract and introduction, when the methods for this research is presented, the authors focus on the 'literature review' that was conducted. It also mentioned the focus and importance given to 'evidence-based literature'. When reading this section, initially, I had questions regarding the cultural appropriatedness of the methods, i.e. a review of the literature when we know that much of this literature continues to be dominated by Western/Non-Indigenous methodologies/ways of seeing and doing research. I think it would be relevant for the authors of the paper to see a bit more about the grey literature, what it included, how it was integrated, found; but also if the 'literature review' conducted was validated or discussed with, e.g. a Indigenous-based committee to ensure it was complete and representative.
The same comment goes for the concept of 'evidence'. Later in the paper, in the final part of the results, the authors mention how it emerges that 'evidence' itself is to be defined/understood using the (diverse) perspective of Indigenous peoples, meaning that what counts as evidence, is recognized as evidence, can (is) different. However, this is absent (that is, such a consideration and recognition in this research project) in the earlier parts of the paper when 'evidence-based' is mentioned. I suggest revising, i.e. making sure that the concept of 'evidence' is defined with Indigenous communities, and that examples are provided.
In the results too, one participant at least mentioned how 'what is published', 'evidence found in publications' is not sufficient or appropriate. However, without further details re: the 'grey literature' in this research, it appears this research would reproduce and commit the same fault reproached by this participation, i.e. a focus on what is 'published' as the current evidence on the topic. I would suggest the authors either (or both) add clarification regarding the 'review of the literature' and how it was compensated/negociated from an Indigenous-based methodological point of view, and, that in the limitations, this is also mentioned, i.e. the focus (perhaps?) on published findings; and perhaps (a suggestion) adding 1-2 recommendations on how this approach/method could be enhanced.
The findings are so rich! It's hard to know where to begin! I do think, however, that some of what is written, in the earliest parts of the paper, should be edited in light of the rich findings. I had some criticisms regarding some of the language (e.g. 'translating evidence for Aboriginal communities'), as is written in the earliest parts; but then, reading the findings, I could see how the authors would have a different views re: some important concepts and how to do well and appropriately/ethically partnership research, i.e. a continuous and self-reflexive approach to cultural appropriately; what counts as 'evidence' or 'methods'; whose capacities should be enhanced; how even seemingly 'positive and emancipatory' langage such as 'enhance capabilities' can harbor a veiled negative viewpoint (e.g. who has and doesn't have capacities/capabilities; who's need to be enhanced, Indigenous or non-Indigenous peoples and communities).
Finally, I do think it would be worthwile, in the discussion or conclusion, to say a few words about the 'larger context and conditions of academic-based research and knowledges', that is: that they find themselves inscribed in an economy of knowledge, based in a Capitalist economic system, meaning that the knowledge becomes a product, an economic product, and that any research finds itself in this concerning place, thus always treatening the capacity to embrace a more ethical goal of producing knowledge to advance in meaningful, effective, ethical and truthful ways the true needs of (Indigenous) communities.
I hope these comments are helpful! Again, I thank the authors for such a paper and research. This is extremely pertinent and helpful. I look forward to the published paper to recommend to colleagues and use myself.
Author Response
Dear Reviewer 1,
We thank you for your careful and considered review of our manuscript, and your valuable advice for minor adjustments to improve the paper. We do suggest that Reviewer 1 is directed to read the companion paper to this manuscript, recently published in this special issue, which presents the comprehensive review methods and findings in detail. We have addressed Reviewer 1 suggestions as follows:
- Provide more emphasis early in the paper about the grey literature
Response: As this manuscript is the second of two companion papers, we feel that the cultural rigor employed by our team in identifying and exploring the existing literature, including grey literature, as well as the collaborative analysis of that comprehensive review, is reported in great detail within the first of the companion papers. We do not feel that further elaboration of these processes relevant to the comprehensive review are needed within this second paper, which instead focuses on the stakeholder enagement and revision of the evidence that was collected via the literature search. We have amended the references to the review to 'comprehensive review' rather than 'comprehensive literature review' to better reflect the scope of the review and align with the terminology in paper one. - Concept of 'evidence' is defined
Response: We do agree with Reviewer 1 that the concept of 'evidence' is a contested space, and this issue was a foundational principle in why the findings of the comprehensive review, were considered, reviewed and revised by the broad set of stakeholders. I have made minor edits to the wording around 'evidence base' in paragraph 3 of the introduction. - Add clarification regarding the 'review of the literature'
Response: Again, the robust approach that was taken by our team regarding the collaborative analysis of the comprehensive review is described in detail in paper one. We assume that Reviewer 1 hasn't read our review companion paper, as we feel that these points of concern are thoroughly addressed in paper one. - Some of what is written, in the earliest parts of the paper, should be edited in light of the rich findings
Response: We greatly appreciate Reviewer 1's sentiments regarding the need to avoid apotential veiled negative viewpoint associated with the phrasing of some aspects of the findings around enhancing capabilities. It is difficult for us to make such central changes to the wording of these findings, as they have been parsed and developed through the extensive Collaborative Yarning Methodology, with the First Nations Co-Design Working Group (FNCDWG) and Cancer Australia’s Aboriginal and Torres Strait Islander Cancer Control Leadership Group (the Leadership Group).
Thanks again for your positive comments and your considered review. We are most grateful!
Regards,
Kate Anderson, on behalf of all authors
Reviewer 2 Report
The author explores a set of key principles and best practices to promote the health of Australian aborigines. These principles and practices may provide a valuable starting point for Australia to develop guidelines and evaluation standards in the future. However, if the research provides a flow chart, it may make it easier for the reader to master the research process. If the research provides some detail in the qualitative research process, the discussion process and effect may be better evaluated.
Author Response
Dear Reviewer 2,
Many thanks for your positive review of our manuscript. We accept your suggestion, and we have added a flow chart at the end of the Methods section on page 5 to depict the research process.
Regards,
Kate Anderson, on behalf of all authors
